



**Environmental behaviors of (*E*)-Pyriminobac-methyl in agricultural**
**soils**
Wenwen Zhou[a], Haoran Jia[b], Lang Liu[b], Baotong Li[b*], Yuqi Li[b], Meizhu Gao[b]
[a] College of Food Science and Engineering, Jiangxi Agricultural University, Nanchang 330045, China
[b] College of Land Resources and Environment, Jiangxi Agricultural University, Nanchang 330045, China
*Corresponding Author: Baotong Li, Tel.: 86-791-83813420, E-mail: btli666@163.com



**Abstract**
(*E*)-Pyriminobac-methyl (EPM), a pyrimidine benzoic acid esters herbicide, has a high potential as
weedicide; nevertheless, its environmental behaviors are still not well understood. In this study, we
systematically investigated for the first time the adsorption–desorption, degradation, and leaching
behaviors of EPM in agricultural soils from five exemplar sites in China (characterized by different
physicochemical properties) through laboratory simulation experiments. The EPM adsorption–desorption
results were well fitted by the Freundlich model ($R^2 > 0.9999$). In the analyzed soils, the Freundlich
adsorption (i.e., $K_{f\text{-}ads}$) and desorption (i.e., $K_{f\text{-}des}$) coefficients of EPM varied between 0.85–32.22 $mg^{1-1/n}$
$L^{1/n}$ $kg^{-1}$ and between 0.78–5.02 $mg^{1-1/n}$ $L^{1/n}$ $kg^{-1}$, respectively. Moreover, the degradation of EPM
reflected first-order kinetics: its half-life ranged between 37.46–66.00 d depending on the environmental
conditions, and abiotic degradation was predominant in the degradation of this compound. The mobility of
EPM in the five soils varied from immobile to highly mobile. The groundwater ubiquity score ranged
between 0.9765–2.7160, indicating that EPM posed threat to groundwater quality. Overall, the results of
this study demonstrate the easy degradability of EPM, as well as its high adsorption affinity and low
mobility in soils with abundant organic matter content and high cation exchange capacity. Under such
conditions, there is a relatively low contamination risk for groundwater systems in relation to this
compound. At the same time, due to its slow degradation, EPM has a low adsorption affinity and tends to
be highly mobile in soils poor in organic matter content and with low cation exchange capacity. Under
such conditions, there is a relatively high contamination risk for groundwater systems in relation to this
compound. Overall, our findings provide a solid basis for predicting the environmental impacts of EPM.
**Keywords:** (*E*)-pyriminobac-methyl, adsorption–desorption, degradation, leaching, agricultural soils



## 1. Introduction

Herbicides are usually applied to chemically control the growth of weeds associated with different types of crops, both in China and worldwide (Barchanska et al., 2021; Brillas, 2021). Unfortunately, with the applications of weedicides, they have been detected outside of their original application sites, meaning that they contribute to environmental contamination (Jiang et al., 2018; Perotti et al., 2020). In recent years, the groundwater pollution caused by herbicides has attracted increasing attention worldwide (Khan et al., 2020; Wu et al., 2017). Importantly, the environmental fate of herbicides in soil mainly depends on the adsorption–desorption, degradation, and leaching processes. In fact, herbicides can be transferred from soil to groundwater through surface runoff or leaching, resulting in groundwater pollution (Cueff et al., 2020; Gawel et al., 2020). Furthermore, the adsorption–desorption rate and the degradation capability of herbicides regulate the migration of herbicides: the groundwater ubiquity score (GUS) can be used to evaluate their ecological and environmental safety (Acharya et al., 2020; Liu et al., 2021). However, few scholars have assessed the effects of soil properties on the adsorption–desorption, degradation, and leaching behaviors of weedicide, especially the environmental consequences of these changes.

Pyriminobac-methyl (PM)[methyl-2-(4,6-dimethoxy-2-pyrimidinyloxy)-6-(1-methoxyiminoethyl) benzoate] (Fig. S1), is composed of a mixture of its (*E*) - isomer (I) and (*Z*) - isomer (II) as the active ingredient due to its chemical structure contain oxime(Song et al., 2010), a mixture of two isomers (I and II) in a > 9:1 (major/minor) ratio which was developed from sulfonylurea by Kumiai Chemical Industry Co., Ltd. In 1996 (Tokyo, Japan)(Tamaru and Saito, 1996). Tamaru et al. (1997)) reported that (*E*) - isomer (I) has been confirmed to restrain the plant enzyme acetolactate synthase (ALS) and prevent branched chain amino acid biosynthesis, and the (*E*) - pyriminobac-methyl (EPM) showed stronger soil adsorption and weaker hydrophilic properties





than (*Z*) - pyriminobac-methyl (ZPM), thus EPM was selected as the best compound to develop a commercial
weedicide, which is commonly used to control the growth of sedges and both gramineous and annual weeds.
The chemistry of EPM is well understood; the octanol-water partition coefficient is 2.31 (low) at pH 7,
20 °C, the solubility - in water is 9.25 mg L$^{-1}$ (low) at 20 °C, and the vapour pressure is just $3.1×10^{-5}$ Pa
(low) at 20 °C (Lewis et al., 2016).   A distinct advantage of EPM as a weedicide is that, this compound has
an herbicidal activity 1.5–2 times higher and requires an application rate 1/5–1/10 lower than
bensulfuron-methyl (a broad-spectrum herbicide) on *Echinochloa crusgalli* and *Leptochloa chinensis* (Iwakami
et al., 2015; Shibayama, 2001; Song et al., 2010). Notably, EPM can prevent the growth of *E. crusgalli* and *L.*
*chinensis* populations and suppress them effectively over long periods, while being non-toxic, and eventually
increasing the yield of paddy rice and subsequent crops (e.g., rape, cabbage, *Astragalus smicus*, wheat, and
potato) (Iwafune et al., 2010; Qin et al., 2017; Tang et al., 2010; Yoshii et al., 2020). Nevertheless, few studies
have lucubrated the environmental behaviors of EPM after it was widely used as herbicide in the farming
industry.

Most former investigations on EPM as a weedicide mainly focused on the photo-transformation in water

and low temperature storage stability in paddy rice. Inao et al. (2009)) demonstrated that the photoconversion of
PM in water is the main fate, and the main process is EPM / ZPM reached approximately equilibrium after 4.5 h,
furthermore, the EPM / ZPM ratio is about 1/1.35. Another researcher found that even if proper water
management to prevent EPM surface runoff from paddy fields was practiced, a significant amount of EPM
components were discharged into drainage channels through percolation (Sudo et al., 2018). Nevertheless, the
effects of soil properties on the adsorption–desorption, degradation, and leaching behaviors of EPM have rarely
been reported.





A number of researchers have reported that the soil matrix is a highly complicated system, in which
environmental processes (e.g., the sorption–desorption and leaching of herbicides) are affected by multiple
factors, including the soil organic matter (OM) content, pH, cation exchange capacity (CEC), microbial or
chemical degradation, chemical type, environmental conditions (e.g., temperature, humidity, and rainfall), and
texture (Alonso et al., 2011; Rao et al., 2020; Xie et al., 2020; Zhou et al., 2019a). Nevertheless, soil organic
or inorganic colloids and pH (pH < $pK_a$ neutral state and pH > $pK_a$ negative charge) can influence soil–herbicide
interactions. In this context, the leaching of anionic compounds is likely (Pérez-Lucas et al., 2020). Moreover,
the leaching of herbicides in soil and the associated risk of water pollution are both affected by sorption and
desorption (Xie et al., 2020).
Until present, the environmental fate of EPM in soils has not been studied in detail. Clarifying the
adsorption and transport of EPM in soil is very important for the protection of surface water and groundwater
from EPM pollution. Hence, this study aimed at: 1) gaining an essential understanding of the
adsorption–desorption, degradation, and leaching behaviors of EPM in agricultural soils through laboratory
simulation experiments; 2) determining the effects of soil properties on the above behaviors in agricultural soils;
and 3) conducting a basic evaluation of the safety and applicability of EPM in the environment. Overall, our
results provide a scientific basis for the prevention or, at least, minimization of the possible effects of EPM on
groundwater, as well as for modeling the fate of EPM in the environment and the potentially associated risks.
**2. Materials and Methods**
*2.1. Chemicals*



EPM (99.0%; chemical formula: $C_{17}H_{19}N_3O_6$; structure shown in Fig. S1) was obtained from ZZBIO Co.,
Ltd. (Shanghai, China). Moreover, we used only organic solvents of chromatographic grade (Sigma-Aldrich,
Germany). EPM was dissolved in acetonitrile, obtaining a 1000 mg $L^{-1}$ test mother liquor. Moreover, a standard
EPM working solution (0.01–5.00 mg $L^{-1}$) was prepared by diluting the stock solution with a $CaCl_2$ solution
(0.01 mol $L^{-1}$), which was used as an electrolyte to maintain a constant ionic strength and reduce the cationic
exchange.
In March 2020, five different soils were sampled from the surface layer (0–20 cm) of paddy fields located
in five Chinese provinces: Phaeozem (S1, from Heilongjiang), Anthrosol (S2, from Zhejiang), Ferralsol (S3,
from Jiangxi), Alisol (S4, from Hubei), and Plinthosol (S5, from Hainan). The soil samples were all air-dried,
ground, and passed through a 2-mm sieve before being used. Afterward, standard soil testing methods were
applied to define the basic physicochemical properties of the soils (Table S1) (Gee, 1986; Jackson, 1958;
Nelson, 1985), which were then classified based on the system of the World Reference Base for Soil Resources
(WRB) (L'huillier, 1998). Interestingly, the EPM residues in the analyzed soils were always below the detection
limit.
*2.2. Extraction and final analyses*
The soil samples were transferred to centrifuge tubes and 10 mL of acetonitrile (containing 0.1% of ammonia
water) were added to each of them for extracting EPM. After vortexing the tubes for 5 min, we added 2 g of
NaCl and 3 g of $MgSO_4$. Then, the tubes were capped and vortexed again for 1 min and centrifuged at 2,400 × g
for 5 min. The supernatant (1.5 mL) was transferred into a 2.5-mL single-use centrifuge tube that was already
containing the sorbent (50 mg $C_{18}$ + 150 mg $MgSO_4$). Afterward, all the samples were vortexed again for 1 min
and centrifuged at 5,000 rpm for 5 min. Finally, the resulting supernatant was extracted with a sterile syringe,



passed through a 0.22-μm organic membrane filter, and poured into vials for UPLC system (1260 series, Agilent
Technologies, USA) equipped with a triple quadrupole mass spectrometer (6460C series, Agilent Technologies)
using positive ion mode in multiple reaction monitoring (MRM) mode analysis. The instrument parameters for
Agilent 6460C QQQ UPLC-MS/MS analysis are as follows: The flow rate was maintained at 0.2 mL min$^{-1}$, and
the column (Agilent ZORBAX Eclipse XDB-C18, length 150 mm, inner diameter = 4.6 mm, 5μm coating) was
heated to 35°C. The mobile phase A was water which consisted of 0.1% formate and mobile phase B was
acetonitrile. Gradient condition was: 0.0-0.5 min, 20% B; 0.5-1.0 min, 20%-80% B; 1.0-4.0 min, 80% B; 4.0-5.0
min, 20% B. The mass spectrometer was operated in electrospray ionization positive with MRM scanning mode,
dry gas temperature at 500 °C, Ion source temperature at 150 °C, desolvation gas flow at 1000 L h$^{-1}$; capillary
voltage at 2500 V; cone voltage at 18 V and collision gas was argon, dwell time at 50 ms, collision pressure at
58 eV.

The efficiency of the EPM extraction during the adsorption–desorption, degradation, and leaching

experiments was evaluated based on the results of recovery experiments. The average recovery rates of EPM in
the adsorption–desorption experiments, at initial spiked concentrations of 0.1 and 1.0 mg kg$^{-1}$ in the soils, varied
between 94.3–102.4% (relative standard deviation (RSD) = 1.1–3.8%). Meanwhile, the average recovery rates
of EPM in soil in the degradation experiments, at initial spiked concentrations of 0.01, 0.2, and 2.0 mg kg$^{-1}$ in
the soils, ranged between 92.6–106.0% (RSD = 1.1–2.9%). Furthermore, the average recovery rates of EPM at
initial spiked concentrations of 0.0001, 0.01, and 0.1 mg L$^{-1}$ in the supernatant of soils were 88.7–107.9% (RSD
= 1.7–4.9%). Furthermore, the average recovery rates of EPM in the leaching experiments at initial spiked
concentrations of 0.05 and 1.0 mg kg$^{-1}$ in the soils were 95.8–109% (RSD = 1.6–4.4%).
*2.3. Soils samples*



The batch equilibration method suggested by the GB 31270.4-2014 guidelines: Adsorption/Desorption in
Soils for these soils (Gb, 2014b) was applied to conduct adsorption–desorption experiments. First, for the
adsorption kinetics tests, each soil sample (2.0 g) was introduced in a centrifuge tube containing 10 mL of a
EPM aqueous solution (1 mg L$^{-1}$). For each of these tubes, we also analyzed a blank tube (which contained no
herbicide) and a control tube (which contained no soil). All the tubes were then shaken by an oscillator at 25 °C
± 1 °C for different time intervals of 0.5, 1, 2, 4, 6, 8, 12, 16, 20, and 24 h.
The desorption kinetics were analyzed instead by taking 5 mL of supernatant from each tube after
adsorption equilibration and by replacing them with an equal volume of CaCl$_2$ solution (which contained no
EPM). A microvortex mixer was used to thoroughly mix the resulting solution and an oscillator was used to
shake it at 25 °C ± 1 °C for several time intervals: 0.5, 1, 2, 4, 6, 8, 12, and 24 h. Finally, for the
high-performance liquid chromatography-mass spectrometry (UPLC-MS/MS) analyses, the samples were
centrifuged for 10 min at 2,400×g and the supernatants were filtered through 0.22-μm mixed-cellulose ester
filter membranes.
The adsorption–desorption equilibrium time of EPM in the five soils was 24 h (Fig. 1); moreover, the
initial EPM concentrations adopted for these experiments were 0.01, 0.10, 0.50, 1.00, and 5.00 mg L$^{-1}$. The
concentration of EPM in the supernatant was determined after centrifugation. Then, the amount of
adsorbed–desorbed EPM in each soil was calculated based on the concentration of EPM in the solution before
and after the adsorption–desorption process. The supernatant removed after the adsorption experiments was
replaced with 5 mL of CaCl$_2$ containing no EPM; then, the tubes were shaken for 24 h and centrifuged. Finally,
the EPM concentration was determined based on the supernatant collected after this procedure. Considering the





results of preliminary experiments and with the aim of desorbing the majority of the adsorbed EPM, we decided
to repeat the desorption process for at least three times.
*2.4. Degradation experiments*
By following the GB 31270.1-2014 guidelines(Gb, 2014c), we performed a series of EPM soil degradation
experiments. To ensure aerobic conditions, 20 g of each type of agricultural soil were weighed and introduced
in 250-mL Erlenmeyer flasks (in three replicates). Ultrapure water was added during the subsequent cultivation
process in order to maintain the soil water content at 60% of the maximum water holding capacity. We then
spiked each soil sample with 400 μL of the 100 mg $L^{-1}$ EPM working solution (achieving an initial
concentration of 2 mg $kg^{-1}$ in the soil: the water-soluble, organic solvent volume was ≤1%) and then cultured
in the dark in an incubator kept at 25 ± 1 °C. Subsequently, we collected three parallel sub-samples on 0, 1, 2,
4, 6, 10, 15, 30, 45, 60, 90, and 120 day, and the EPM content was determined by UPLC-MS/MS on the
respective days of collection. The amount of water in the Erlenmeyer flasks was periodically adjusted during
the culturing process with the aim of retaining the original water-holding state. Each treatment was done in
triplicate, totalizing 60 samples per treatment (5 soil samples per treatment per sampling day; 12 sampling days
in total), The following experiment was done in the same way.
Another set of experiments was conducted under anaerobic conditions. In this case, we first cultured the
soil samples for 30 days and then added a 2 cm-thick water layer to each of them. To maintain the desired
conditions, $N_2$ was continuously introduced into the culture system. The soil samples were subsequently moved
into an incubator and cultivated in the dark at 25 ± 1 °C. Finally, three parallel sub-samples were collected on 0,
1, 2, 4, 6, 10, 15, 30, 45, 60, 90, and 120 day, and the EPM content was determined by UPLC-MS/MS on the
respective days of collection.



A set of degradation experiments was performed under sterilized conditions. With this objective, the
sterilized soils (20 g each) were weighed and introduced in 250-mL Erlenmeyer flasks in three replicates.
Notably, in order to keep the soil water content at 60% of the maximum water holding capacity, sterile water
was added during the cultivation process. Then, each soil sample was spiked with 400 µL of the 100 mg L$^{-1}$
EPM working solution, achieving an initial concentration of 2 mg kg$^{-1}$ (the water-soluble, organic solvent
volume was ≤1%). The samples were hence moved into an incubator and cultured in the dark at 25 ± 1 °C.
Three parallel sub-samples were collected on 0, 1, 2, 4, 6, 10, 15, 30, 45, 60, 90, and 120 day, and the EPM
content was determined by UPLC-MS/MS on the respective days of collection.
These experiments were done under different soil moisture conditions and aerobic conditions, at a EPM
fortification level of 2 mg kg$^{-1}$. After adjusting their moisture by adding water (water percentage = 40%, 60%,
and 80% of the total volume), the soils were incubated in the dark at 25 ± 11 °C. During this last phase, we
regularly added ultrapure water to keep the moisture at 40%, 60%, and 80%.
*2.5. Leaching experiments*
The herbicide leaching process was investigated by following the GB 31270.5-2014 guidelines (Gb, 2014a).
PVC columns (length = 35 cm, internal diameter = 4.5 cm), each hand-packed with 600–800 g of one soil type,
were used to observe the downward movement of the herbicide. Notably, the top 3 cm and the bottom 2 cm
were filled with quartz sand (for minimizing soil disturbance) and glass wool + sea sand (for avoiding soil loss).
After packing each column, we removed any air still present in the column by adding 0.01 mol L$^{-1}$ CaCl$_2$;
moreover, the excess water was eliminated by gravity. The pore volume (PV) was determined by subtracting the
volume of water leached from that of the water added. Subsequently, 1 mL of acetonitrile solution containing
200 µg mL$^{-1}$ of the herbicide (spiking level = 1 µg g$^{-1}$) was added to the top of each column. afterward, the



adsorption equilibrium was achieved by infiltrating 700 μL of 100 mg $L^{-1}$ EPM solution into soil surface and
leaving it to rest for 24 h. To simulate rainfall leaching, 2,000 mL of 0.01 mol $L^{-1}$ $CaCl_2$ solution (21 mL $h^{-1}$)
were added into the soil column at a peristaltic pump speed of 250 mL 12 $h^{-1}$. The leachate was collected every
8 h with a conical flask. Subsequently, each soil column was extracted, cut into three parts (length = 10 cm), and
analyzed by UPLC-MS/MS on the same day. The total mass of the leachate and soil fractions along the soil
column was determined, together with the EPM and water contents within each of them.
*2.6. Data analysis*

The relationship between the concentrations of EPM sorbed in the soil and in the aqueous solution during

the sorption–desorption equilibrium was described through the linear [Eq. (1)] and Freundlich [Eq. (2)] models
(Azizian et al., 2007; Yang et al., 2021):

Linear model: $C_s = KC_e + C$                               (1)

Freundlich model: $C_s = K_f C_e^{1/n}$                        (2)

where $C_s$ (mg $kg^{-1}$) indicates the adsorption of EPM in the soil, $C_e$ (mg $L^{-1}$) the EPM concentration in the
solution during the adsorption equilibrium, C (mg $kg^{-1}$) the amount of soil adsorption when the EPM
concentration was 0 during the adsorption equilibrium, $K$ (mL $g^{-1}$) and $K_f$ ($mg^{1-1/n}$ $L^{1/n}$ $kg^{-1}$) the
adsorption–desorption constants of the linear and Freundlich models, respectively ($K_{f-ads}/K_{f-des}$ in the
adsorption–desorption process), and 1/n the adsorption empirical constant (which provides information about
the non-uniformity of the adsorbent surface).

For the isothermal sorption tests, the amount of EPM adsorbed in the soil was estimated using the subtractive

method [Eq. (3)]:

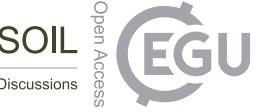

$$C_s = \frac{(C_0 - C_e) \times V}{m}$$


(3)

where $C_0$ (mg L$^{-1}$) is the amount of soil adsorption when the concentration of EPM was 0 during the adsorption
equilibrium, m the soil mass (2.0 g), and $V$ the solution volume (10 mL).
The amount of EPM retained by the soil after desorption was obtained instead by using [Eq. (4)], while the
hysteresis index ($H$) was estimated by applying [Eq. (5)] (Fan et al., 2021; Zhang et al., 2020b):

$$C_{sj} = \frac{C_0 \times V}{m} - \frac{C_{ej} \times V}{2m} - \frac{V}{m} \sum_{n=1}^{j} C_e (j-1)$$


(4)

$$H = \frac{1/n_{des}}{1/n_{ads}}$$


(5)

where $C_{sj}$ (mg kg$^{-1}$) is the concentration of EPM adsorbed by the soil after the j-th desorption (i = 1–5), $C_{ej}$
(mg L$^{-1}$) the EPM concentration in the supernatant after the j-th desorption, $H$ the hysteresis coefficient, and
$1/n_{ads}$ and $1/n_{des}$ the empirical adsorption and desorption constants, respectively.
The distribution coefficient ($K_d$) was calculated based on the distribution ratio of EPM in the water–soil
system by using [Eq. (6)] (Carballa et al., 2008; Ternes et al., 2004):

$$K_d = \frac{C_s}{C_e}$$


(6)

The sorption constants of the OC ($K_{OC}$) and OM ($K_{OM}$) contents were calculated through [Eqs. (7) and (8)]
(Rae et al., 1998; Zhang et al., 2011), respectively. Moreover, the Gibbs free energy change of sorption ($\Delta G$, kJ
mol$^{-1}$) (Jia et al., 2019) and the GUS (Gustafson, 1989) were calculated as follows:

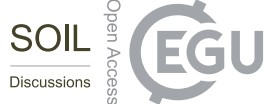

$$K_{OM} = 100 \times K_{f-ads} / OM\%$$
(7)

$$K_{OC} = 100 \times K_d / OC\%$$
(8)

$$\Delta G = -RT \ln K_{OM} / 1000$$
(9)

$$GUS = \lg t_{1/2} \times (4 - \lg K_{OC})$$
(10)

where $OM$ % and $OC$ % represent the soil OM and OC contents, respectively, $R$ the molar gas constant
(8.314 J K$^{-1}$ mol$^{-1}$), $T$ (K) the absolute temperature, and $t_{1/2}$ the half-life (in days) given by [Eq. (12)]. Organic
contaminants were categorized into five types: highly adsorbed compounds ($K_{OC} > 20{,}000$), sub-highly adsorbed
compounds ($5{,}000 < K_{OC} \leq 20{,}000$), medium-adsorbed compounds ($1{,}000 < K_{OC} \leq 5{,}000$), sub-difficultly
adsorbed compounds ($200 < K_{OC} \leq 1{,}000$), and difficultly adsorbed compounds ($K_{OC} \leq 200$)(Gb, 2014b).
The degradation data relative to herbicides in soil could be successfully fitted to a first-order kinetic model
[Eq. (11)], previously used in similar studies (Bailey et al., 1968; Liu et al., 2021; Ou et al., 2020):
$$C_t = C_0 e^{-kt}$$
(11)

where $C_t$ (mg kg$^{-1}$) and $C_0$ (mg kg$^{-1}$) are the concentrations of EPM in the soil at incubation times t (d) and
0 (d), respectively, while $k$ is the first-order rate constant (d$^{-1}$).
The half-life ($t_{1/2}$) to be used in above model was calculated through [Eq. (12)] (Yin and Zelenay, 2018):
$$t_{1/2} = 0.693/k$$
(12)

Four categories of herbicide degradability were defined: easily degradable ($t_{1/2} \leq 30$), moderately
degradable ($30 < t_{1/2} \leq 90$), slightly degradable ($90 < t_{1/2} \leq 180$), and poorly degradable ($t_{1/2} > 180$)(Gb, 2014c).





Based on the content of EPM in different sections of the soil columns and in the leachate [Eq. (13)](Gb,
2014a), we were able to calculate the leaching rate of EPM:
$$R_i = \frac{m_i}{m_0} \times 100$$  (13)
where $R_i$ (%) is the ratio of EPM content in each soil section or in the leachate to the total added amount, $m_i$
(mg) the mass of EPM in each soil section (where $i$ = 1, 2, 3, and 4, representing the 0–10 cm, 10–20 cm, and
20–30 cm soil sections and in the leachate, respectively), and $m_0$ (mg) the total added amount of EPM ($m_0$ =
0.02 mg). Regarding the mobility scheme we defined the following $R_i$ ranges: class 1 (immobile, $R_1 > 50$ %),
class 2 (slightly mobile, $R_2 + R_3 + R_4 > 50$ %), class 3 (mobile, $R_3 + R_4 > 50$ %), and class 4 (highly mobile,
$R_4 > 50$ %)(Gb, 2014a).
The data fittings (to the linear and Freundlich models for the adsorption isotherms and to the simple
first-order kinetic model for degradation) were conducted with OriginPro 8.05 (OriginLab Corp., Northampton,
USA). All the values reported here were calculated as the means of three replicates; furthermore, the differences
between these means were statistically analyzed through Duncan's multiple range test, while their reciprocal
relationships were determined though a Spearman's correlation analysis using SPSS Statistics 22.0 (IBM SPSS,
Somers, USA).
**3. Results and discussion**
*3.1. Adsorption–desorption kinetics*
The adsorption and desorption kinetic curves of EPM in different types of agricultural soils are shown in
Fig. 1. After EPM had been in contact with the soil solution for 1 h, the concentration of EPM exhibited a sharp
drop (from 0 to 95.35, 75.45, 51.57, 77.41 and finally 65.84 % between S1–S5). This event corresponded to the



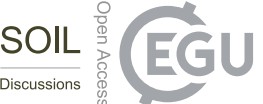

fast sorption phase. After 2–8 h, the EPM soil system entered the slow adsorption stage and there was a gradual
increase in the sorption of EPM. This last process reached an equilibrium state of EPM sorption after 8 h, which
was reflected by stable concentrations of EPM. The sorption of EPM decreased from the Phaeozem (S1, 97.99%)
to the Anthrosol (S2, 79.69%), Alisol (S4, 77.81%), Plinthosol (S5, 72.57%), and Ferralsol (S3, 52.35%) (Fig.
1a). This trend reflected the soils' OM contents. Previous studies have also found that the sorption of organic
chemicals in soils is mainly related to their OM contents (Xu et al., 2021; Zhou et al., 2019b).
The desorption equilibration of EPM in soil was slightly slower and a hysteresis effect was observed. The
rapid and slow desorption stages occurred between 0–2 h and 2–12 h, respectively; afterward, the concentration
of EPM remained unchanged, until the desorption process reached its equilibrium state (within 24 h). Based on
these data, we defined 24 h as the period of EPM adsorption-desorption. The desorption value of EPM observed
in our experiments after 24 h increased from the Phaeozem (S1, 8.04%) to the Anthrosol (S2, 12.07%), Alisol
(S4, 14.48%), Plinthosol (S5, 17.55%), and Ferralsol (S3, 24.08%) (Fig. 1b).
The sorption of OM in soil typically occurs during the rapid reaction and slow equilibrium phases (Calvet,
1989). Therefore, the reduction of the EPM content in the solution before and after the experiment was likely
due to soil sorption. According to the above results, the soil sorption rate was inversely proportional to the soil
desorption rate toward EPM.
*3.2. Adsorption–desorption isotherms*
Non-linear adsorption–desorption isotherms of EPM were observed (Fig. 1). When the concentration of
EPM was low, this compound was preferentially adsorbed by OM (which has a strong adsorption capacity);
meanwhile, soils with higher OM contents (e.g., Phaeozems, S1) desorbed EPM slowly. The positive





292 relationship between sorption and OM has been reported previously (Hochman et al., 2021; Obregón

293 Alvarez et al., 2021; Patel et al., 2021). Moreover, the adsorption ability of EPM has been found to be high,

294 similar to those of other herbicides (e.g., chlorsulfuron, imazamethabenz-methyl, flumetsulam, and

295 bispyribac-sodium) (Kalsi and Kaur, 2019; Medo et al., 2020; Spadotto et al., 2020). Generally, a low

296 mobility of herbicides in soil is related to a high sorption constant. Hence, the EPM contained in the soils

297 tested in this study (excluding the phaeozem, S1) is likely to have been polluting the groundwater and

298 surface water of the respective areas of origin.

299 OM adsorption in soil is currently explained mainly by partitioning and adsorption-site theories (Martins and

300 Mermoud, 1998), which are well described by the linear and Freundlich isotherm models, respectively. Our

301 isothermal sorption and desorption data were thus fitted to these two models: the obtained fitting parameters are

302 listed in Table 1. The average $R^2$ value for the linear model (0.9950) was smaller than that for the Freundlich

303 model (0.9999); moreover, the $C$ values obtained for the Plinthosol (S5, –0.01 ± 0.06) by fitting the data to the

304 linear model were negative (Table 1) and did not meet the experimental requirements, indicating that this type of

305 model was not suitable for this experiment. Meanwhile, the sorption-site theory was found to more accurately

306 describe the sorption–desorption process: the Freundlich model provided a more accurate description of the

307 EPM sorption-desorption characteristics observed in this study.

308 Generally, larger $K_{f\text{-}ads}$ values correspond to higher sorption capacities (Carneiro et al., 2020; Khorram et al.,

309 2018; Silva et al., 2019). Here, the $K_{f\text{-}ads}$ values of EPM ranged between 0.85 (in S4) and 32.22 (in S1) (mg$^{1-1/n}$

310 L$^{1/n}$ kg$^{-1}$), while the $1/n_{f\text{-}ads}$ values ranged between 0.80 (S1) and 1.06 (S5) (Table 1). In brief, S5 showed an

311 S-type adsorption isotherm (since $1/n_{f\text{-}ads} > 1$), while S1, S2, S3, and S4 showed an L-type adsorption isotherm

312 (since $1/n_{f\text{-}ads} < 1$). In this study, the $H$ values of EPM ranged between 0.013 (Phaeozem, S1) and 0.845



(Ferralsol, S3). Since the *H* values were < 0.7 in S1, S2, S4, and S5, these particular soils showed a positive
hysteresis: the desorption rate of EPM was lower than its sorption rate. Meanwhile, since the *H* values in S3
were between 0.7–1.0, the sorption and desorption rates were in equilibrium: S3 did not exhibit any obvious
hysteresis. Similar results were reported that hysteresis was absent when $0.7 < H < 1$(Gao and Jiang, 2010; Yue
et al., 2017; Barriuso et al., 1994).

Soil physicochemical properties are important factors influencing herbicide adsorption behaviors (Urach

Ferreira et al., 2020; Wei et al., 2020). We determined the relationship between the Freundlich
adsorption–desorption constant and the soil physicochemical (soil pH, CEC, soil clay content, OM content, and
OC content) properties and carried out a linear correlation analysis based on the experimental data fitting (Table
S2). The results showed that the soil pH, CEC, soil clay content, OM content, and OC content were positively
correlated with $K_{f\text{-}des}$ and $K_{f\text{-}ads}$ (slope > 0). In soils, some polar contents, ionizable groups, and the CEC tend to
increase during OM humification (Calvet, 1989; Meimaroglou and Mouzakis, 2019; Rae et al., 1998). This
mechanism possibly explains the adsorption of EPM in soils high in OM and CEC. Our findings agree with
those of Acharya et al. (2020)) and García-Delgado et al. (2020)): the soil humic acid and clay fractions (high in
OM and CEC and possessing a high number of active sites) are capable of intense EPM adsorption; in contrast,
the soil coarse sand fraction (low in OM and CEC) is characterized by a weaker EPM adsorption. Notably, the
soil with the highest fumigant adsorption capacity was also possibly that with the highest OM abundance and
CEC. For example, strong linear and positive correlations have been found between the adsorption–desorption
of benzobicyclon hydrolysate and the soil clay content, OC content, OM content, and CEC, while moderate
linear and negative correlations were observed between those processes and the soil pH (Rao et al., 2020).

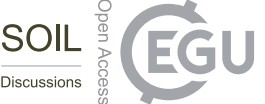

The $K_{OC}$ value is typically used to indicate the EPM sorption capacity of a soil (Fao, 2000; Xiang, 2019)
(see Table 2). EPM was sub-difficultly adsorbed in S2, S3, S4, and S5: this aspect was reflected by the $K_{OC}$
values, which ranged between 200–1,000. However, in S1 the $K_{OC}$ values ranged between 1,000–5,000,
indicating a medium adsorbance of EPM in this soil. Overall, an increasing trend in the mobility of EPM was
observed from the Phaeozem (S1) to the Anthrosol (S2), Alisol (S4), Plinthosol (S5), and Ferralsol (S3). We
hence infer that a relatively low soil adsorption capacity is linked to a relatively high mobility of EPM in that
soil.
The degree of spontaneity of the adsorption process can be quantitatively evaluated based on variations in
the $\Delta G$ values: negative $\Delta G$ values generally indicate that an adsorption process is spontaneous and exothermic
(Nandi et al., 2009). Notably, the change of free energy linked to physical adsorption is smaller than that linked
to chemisorption. The former is in the range of −20 to 0 kJ mol$^{-1}$, while the latter is in the range of −80 to −400
kJ mol$^{-1}$ (Bulut and Aydın, 2006; Yu et al., 2004). We found that the $\Delta G$ values relative to EPM adsorption in all
soils were comprised between −16.2242 and −12.5753 kJ mol$^{-1}$. Therefore, the adsorptions we observed in our
experiments can be regarded as typically spontaneous and exothermic physical adsorptions (Table 2).
*3.3. Degradation of EPM in soil*
To investigate the effects of aerobic and anaerobic microorganisms on EPM degradation, we sterilized the soil
samples or removed all aerobic microorganisms. The soil samples were kept in the dark at 25 °C, maintaining a
soil moisture of 60%. The degradation kinetics of EPM under aerobic, anaerobic, and sterilized conditions are
depicted in Fig. 2, while the fitted parameters are summarized in Table 3. The $R^2$ values for EPM in the five
soils ranged between 0.9313–0.9924, suggesting that the first-order kinetic model agreed with the
correspondent degradation data. The half-life of EPM ranged between 37.46–58.25 d in the aerobic soils,



between 41.75–59.74 d in the anaerobic soils, and between 60.87–66.00 d in the sterilized soils. A moderate
degradation (30 d $< t_{1/2} \leq$ 90 d) of EPM was observed under aerobic, anaerobic, and sterilized conditions. These
results can be partly explained by aerobic and anaerobic transformations occurring in the soils, which have been
described by the GB 31270.1-2014 guidelines for the testing of chemicals(Gb, 2014c). Overall, the half-life of
EPM decreased from the aerobic to the anaerobic and sterilized soils. Understanding the degradation kinetics of
herbicides is critical for predicting their persistence in soil and the soil parameters, which affect regional
agronomic and environmental practices (Buerge et al., 2019; Buttiglieri et al., 2009). Under dark conditions, the
degradation of herbicides in soil mainly results from microbial and abiotic degradation (Marín-Benito et al.,
2019). In this study, when EPM was retained under dark conditions for 30 d, its degradation rates in all soils
under sterilized conditions (35.44, 36.27, 33.27, 32.80, and 34.78%) were a little slower than under anaerobic
(48.60, 41.51, 35.92, 35.61, and 38.07%) and aerobic conditions (53.32, 43.20, 36.73, 35.61, and 39.31%) (Fig.
2). As the degradation rate increased only by 10% compared to that observed under sterilized conditions,
degradation under aerobic/anaerobic conditions appeared to be mainly abiotic degradation. In contrast, other
studies have found that anaerobic microorganisms are predominant contributors in the degradation process and
capable of accelerating it. For example, the degradation rates of phenazine-1-carboxamide (PCN) were much
higher under anaerobic than aerobic conditions, due to its own structural characteristics (Ou et al., 2020).
Between 30–120 d, there were no significant differences in the degradation rates of EPM between sterilized and
unsterilized soils, suggesting that EPM degradation was largely abiotic in this time interval. This might be
attributed to a low bioavailability of EPM for microbial degradation, derived from a high adsorption affinity of
this compound under the right OM content and pH conditions (Liu et al., 2021; Wang et al., 2020a). Overall, it
appears that EPM decomposition in the tested soils was mainly driven by abiotic degradation.





The degradation rate of EPM decreased from S1 to S2, S4, S5, and S3 under both aerobic and anaerobic

conditions (Table 3). A negative correlation was noted between the half-life of EPM and the soil OM content

and CEC under aerobic conditions (slope < 0, P < 0.05; $R^2$ = 0.9478 and 0.8022, respectively); besides, a

negative correlation was observed between the half-life of EPM and the soil OM content under aerobic

conditions (slope < 0, P < 0.05, $R^2$ = 0.8983). Notably, an abundance of OM and high CEC result in an increase

of the carbon sources accessible to microorganisms, effectively stimulating their activity (Xu et al., 2020). In the

presence of microorganisms, the particularly high OM and CEC characterizing S1, resulted in the fastest EPM

degradation among those observed in all soils under aerobic and anaerobic conditions. However, under sterilized

conditions, the degradation rate of EPM decreased from S2 to S4, S1, S5, and S3 (Table 3); moreover, the

half-life of EPM and the soil pH exhibited a negative correlation under these same conditions (slope < 0, P <

0.05, $R^2$ = 0.8850; Table S3). The rate of EPM hydrolysis is known to be positively affected by alkaline soil pH.

This relationship explains why, in the presence of elevated hydrolysis and under sterilized conditions, the fastest

degradation behavior among all the tested soils was observed in S2 (which was characterized by the highest pH).

Notably, the highest differences in the degradation rate of EPM were observed under aerobic conditions. In

order to comprehensively evaluate the influence of various factors on this degradation rate, we hence focused on

the analysis of data collected under aerobic conditions.

The data regarding the degradation behavior of EPM in the tested soils (Table 4 and Fig. 2) conform to

first-order kinetics ($R^2$ > 0.8769). The half-life of EPM varied depending on the moisture conditions: it

diminished from soils with a 60% moisture to those with moisture of 80% and 40%. Additionally, after 120 days,

the degradation rates of EPM in soils with a 40% moisture (74.59, 73.93, 69.98, 73.21, and 71.25 for S1–S5,

respectively) were obviously lower than those in soils with 80% (77.55, 75.38, 72.79, 75.44, and 73.62 for



S1–S5, respectively) and 60% (80.04, 77.31, 75.43, 77.78, and 75.77% for S1–S5, respectively) moistures
(Table 4 and Fig. 2d, e). These results show that, when the soil moisture increased from 40% to 60%, the decay
rate of EPM accelerated, possibly due to the stimulation of a degradation pathway (e.g., through aerobic
microorganisms and chemical hydrolysis) linked to the increase in soil moisture (Wang et al., 2014; Liu et al.,
2021). Conversely, EPM showed a slower decay when the soil moisture increased from 60% to 80%. This
phenomenon might have been caused by an increase in sorption, which would have made EPM less bioavailable.
This effect was more or less important according to the predominance of different biotic pathways of
degradation (Bento et al., 2016; García-Valcárcel and Tadeo, 1999).
*3.4. Leaching potential*
In the current study, leaching experiments were performed by using soil columns, with the aim of
simulating the migration of EPM in several agricultural soils. The correspondent results are shown in Fig. 3. It
was found that the fluidity of EPM was lower in $S_1$ than in $S_2$, $S_3$, $S_4$, or $S_5$. Furthermore, the $R_i$ values of this
compound in $S_1$, $S_2$, $S_3$, $S_4$, and $S_5$ were $R_1 = 99$ %, $R_2 + R_3 + R_4 = 55.5$ %, $R_4 = 71.95$ %, $R_2 + R_3 + R_4 = 76$ %,
and $R_2 + R_3 + R_4 = 74$ %, respectively. Based on the Test guidelines on environmental safety assessment for
chemical pesticide-Part 5: Leaching in soil(Gb, 2014a), the mobility of EPM in the soils S1–S5 was categorized
as immobile, slightly mobile, highly mobile, slightly mobile, and slightly mobile, respectively. The soil OM
content was found to be the most important soil property influencing the mobility of molecular herbicides,
followed by the clay content and the CEC. A lower clay content is usually associated with a higher sand content,
a higher proportion of large pores, a smaller specific surface area per soil unit volume, and a lower adsorption
affinity for herbicides, which, overall, result in a greater herbicide mobility (Boyd et al., 1988; De Matos et al.,
2001; Kulshrestha et al., 2004; Temminghoff et al., 1997). We found that a lower soil OM content corresponded

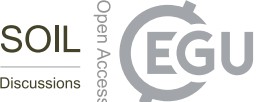

to a weaker adsorption affinity, a weaker tendency of EPM to pass from the soil solution to the solid phase, a
higher availability of EPM for leaching, and a stronger mobility of this same compound. Notably, the OM
content increased from the Ferralsol (S3) to the Plinthosol (S5), Alisol (S4), Anthrosol (S2), and Phaeozems
(S1), while the mobility of EPM increased from the Phaeozem (S1) to the Anthrosol (S2), Alisol (S4), Plinthosol
(S5), and Ferralsol (S3). This mobility tendency is the opposite compared to the adsorption affinity tendency of
EPM in the five soils. As a matter of fact, it is generally known that the mobility of EPM in soil increases as its
adsorption affinity decreases. Similar conclusions were reached through the study of other herbicides (Acharya
et al., 2020; Zhang et al., 2020a).

Here, the GUS was also used to estimate both the leaching potential of chemicals and the risk of

contaminants into groundwater. The GUS values of EPM in S1, S2, S3, S4, and S5 were 0.9765, 2.0402, 2.7160,
2.3755, and 2.6765, respectively (Table 2). The GUS value in S1 was considerably lower than 1.8, EPM should
have little leaching potential in this soil (Gustafson, 1989; Wang et al., 2020b); meanwhile, since the GUS
values in the S2, S3, S4, and S5 soils were between 1.8–2.8, EPM has a considerable leaching potential there
and, possibly, the ability to pollute groundwater (Huang, 2019; Martins et al., 2018). Overall, we can infer that
the risk of groundwater contamination by EPM is low in Phaeozem (S1), due to the low mobility of this
compound; however, the risk is much higher when the same compound is contained in Anthrosol (S2), Ferralsol
(S3), Alisol (S4), and Plinthosol (S5).
**4. Conclusions**
In this study, we found that EPM degrades easily, has a high adsorption affinity and a low mobility in Phaeozem
(S1), which result in a low contamination risk for groundwater systems. On the contrary, this compound
degrades slowly in Anthrosol (S2), Ferralsol (S3), Alisol (S4), and Plinthosol (S5), due to a low adsorption



affinity and moderate mobility, which result in a high contamination risk for groundwater systems. The
adsorption–desorption, degradation, and leaching of EPM were systematically explored in five agricultural soils.
We noticed that physical adsorption was the main mode of EPM adsorption. The effects of soil physicochemical
properties on the adsorption and desorption of this compound were quantified by linear regression analysis. In
this regard, the Freundlich adsorption ($K_{f-ads}$) and desorption ($K_{f-des}$) constants were linearly and positively
correlated with the soil OC content, OM content, and CEC, while nonsignificant correlations were observed
among the above constants and the soil pH and clay content.

The dissipation of EPM depended mainly on soil conditions (i.e., moisture, pH, and soil type). EPM

degradation was most likely derived from abiotic degradation mechanisms; furthermore, the leaching ability of
EPM increased from the Phaeozem (S1) to the Anthrosol (S2), Alisol (S4), Plinthosol (S5), and Ferralsol (S3).
Overall, the high leaching ability and desorption capacity of EPM were accompanied by a low adsorption
capacity and there were no significant relationships between pH and the leaching rate of EPM in the five types
of soils. In contrast, the OM content, CEC, and soil clay content were the main responsible for the observed
leaching rates.

To completely understand the fate of EPM in the environment, it is necessary to perform additional studies

on the microbial community structures and functional diversities of other types of soil besides those analyzed
here. As a matter of fact, there are still only a few studies on the environmental fate of EPM; therefore, our
results may serve as a reference for evaluating the risks involved in the increasingly wide application of this
compound.
**Declaration of Competing Interest**



The authors declare that they have no known competing financial interests or personal relationships that
could have appeared to influence the work reported in this paper.
**Acknowledgements**
This work is financially supported by the National Key Research and Development Plan of China
(2017YFD0301604).

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




**Table 1**
Comparison between the results of the linear and Freundlich models for the adsorption–desorption of EPM in five agricultural soils.

| Soil sample | Soil type | Adsorption | | | | | | Desorption | | | | | |
| | | Linear model | | | Freundlich model | | | Linear model | | Freundlich model | | | |
| | | $K$ (mL g$^{-1}$) [a] | $C_0$ (mg kg$^{-1}$) [a] | $R^2$ | $K_{f\text{-}ads}$ (mg$^{1-1/n}$ L$^{1/n}$ kg$^{-1}$) [a] | $1/n_{ads}$ [a] | $R^2$ | $K$ (mL g$^{-1}$) [a] | $R^2$ | $K_{f\text{-}des}$ (mg$^{1-1/n}$ L$^{1/n}$ kg$^{-1}$) [a] | $1/n_{des}$ [a] | $R^2$ | $H$ |
| --- | --- | --- | --- | --- | --- | --- | --- | --- | --- | --- | --- | --- | --- |
| S1 | Phaeozem | 56.21 ± 3.56 | 0.17 ± 0.01 | 0.9841 | 32.22 ± 4.55 | 0.80 ± 0.07 | 0.9999 | 0.80 ± 0.24 | 0.8384 | 5.02 ± 0.02 | 0.01 ± 33.53 | 0.9999 | 0.013 |
| S2 | Anthrosol | 2.78 ± 0.06 | 0.13 ± 0.04 | 0.9982 | 2.95 ± 0.04 | 0.88 ± 0.03 | 0.9999 | 0.27 ± 0.03 | 0.9823 | 2.27 ± 0.01 | 0.71 ± 0.28 | 0.9999 | 0.807 |
| S3 | Ferralsol | 2.43 ± 0.07 | 0.16 ± 0.05 | 0.9975 | 2.65 ± 0.03 | 0.84 ± 0.03 | 0.9999 | 0.82 ± 0.19 | 0.8988 | 1.73 ± 0.05 | 0.11 ± 1.43 | 0.9999 | 0.131 |
| S4 | Alisol | 0.79 ± 0.01 | 0.05 ± 0.01 | 0.9990 | 0.85 ± 0.02 | 0.95 ± 0.03 | 0.9999 | 0.53 ± 0.05 | 0.9834 | 0.78 ± 0.01 | 0.12 ± 0.01 | 1.0000 | 0.126 |
| S5 | Plinthosol | 2.03 ± 0.07 | −0.01 ± 0.06 | 0.9951 | 1.99 ± 0.05 | 1.06 ± 0.04 | 0.9999 | 2.53 ± 0.18 | 0.9905 | 1.38 ± 0.08 | 0.19 ± 0.56 | 0.9999 | 0.179 |

[a] The values represent means ± standard error (SE, n = 3).





**Table 2**
Empirical constants, Gibbs free energy, and groundwater ubiquity score (GUS) for the adsorption of EPM
in five agricultural soils.

| Soil sample | Soil type | K | $C_e/C_0$ | $K_{f\text{-}ads}$ (mg$^{1-1/n}$ L$^{1/n}$ kg$^{-1}$) | $K_{OC}$ | $K_{OM}$ | $\triangle G$ (kJ mol$^{-1}$) | GUS |
|---|---|---|---|---|---|---|---|---|
| S1 | Phaeozem | 64.4821 | 0.0117 | 32.2230 | 2395.8435 | 695.6897 | −16.2242 | 0.9765 |
| S2 | Anthrosol | 3.0971 | 0.2441 | 2.9540 | 606.7513 | 335.2273 | −14.4143 | 2.0402 |
| S3 | Ferralsol | 2.7861 | 0.2641 | 2.6530 | 289.3500 | 159.6386 | −12.5753 | 2.7160 |
| S4 | Alisol | 0.8393 | 0.5437 | 0.8520 | 413.3906 | 242.8571 | −13.6153 | 2.3755 |
| S5 | Plinthosol | 2.0172 | 0.3314 | 1.9950 | 289.8034 | 165.8333 | −12.6696 | 2.6765 |






















**Table 3**

Degradation kinetic models and parameters of EPM under different conditions.

| Soil sample | Soil type | Aerobic | | | Anaerobic | | | Sterilized | | |
|---|---|---|---|---|---|---|---|---|---|---|
| | | First-order kinetic model | Half-life $t_{1/2}$ (d) | $R^2$ | First-order kinetic model | Half-life $t_{1/2}$ (d) | $R^2$ | First-order kinetic model | Half-life $t_{1/2}$ (d) | $R^2$ |
| S1 | Phaeozem | $C_t = 1.5338e^{-0.0185t}$ | 37.46 | 0.9473 | $C_t = 1.7792e^{-0.0166t}$ | 41.75 | 0.9579 | $C_t = 1.8467e^{-0.0111t}$ | 62.43 | 0.9800 |
| S2 | Anthrosol | $C_t = 1.6419e^{-0.0146t}$ | 47.47 | 0.9707 | $C_t = 1.8599e^{-0.0139t}$ | 49.85 | 0.9696 | $C_t = 1.7543e^{-0.0113t}$ | 60.87 | 0.9551 |
| S3 | Ferralsol | $C_t = 1.9363e^{-0.0119t}$ | 58.25 | 0.9843 | $C_t = 1.9968e^{-0.0116t}$ | 59.74 | 0.9878 | $C_t = 1.9349e^{-0.0105t}$ | 66.00 | 0.9775 |
| S4 | Alisol | $C_t = 1.9476e^{-0.0133t}$ | 52.10 | 0.9924 | $C_t = 1.9477e^{-0.0133t}$ | 52.11 | 0.9924 | $C_t = 1.7086e^{-0.0112t}$ | 61.88 | 0.9313 |
| S5 | Plinthosol | $C_t = 1.7864e^{-0.0126t}$ | 55.00 | 0.9655 | $C_t = 1.9725e^{-0.0121t}$ | 57.27 | 0.9923 | $C_t = 1.8638e^{-0.0109t}$ | 63.58 | 0.9761 |






**Table 4**

Degradation kinetic models and parameters of EPM in soil under different moisture conditions.

| Soil sample | Soil type[a] | Saturation moisture capacity (40%) | | | Saturation moisture capacity (60%) | | | Saturation moisture capacity (80%) | | |
|---|---|---|---|---|---|---|---|---|---|---|
| | | First-order kinetic model | Half-life $t_{1/2}$ (d) | $R^2$ | First-order kinetic model | Half-life $t_{1/2}$ (d) | $R^2$ | First-order kinetic model | Half-life $t_{1/2}$ (d) | $R^2$ |
| S1 | Phaeozem | $C_t = 1.7324e^{-0.0141t}$ | 49.15 | 0.9582 | $C_t = 1.5338e^{-0.0185t}$ | 37.46 | 0.9473 | $C_t = 1.7792e^{-0.0166t}$ | 41.75 | 0.9579 |
| S2 | Anthrosol | $C_t = 1.6551e^{-0.0133t}$ | 52.11 | 0.8769 | $C_t = 1.6419e^{-0.0146t}$ | 47.47 | 0.9707 | $C_t = 1.8599e^{-0.0139t}$ | 49.87 | 0.9696 |
| S3 | Ferralsol | $C_t = 1.8659e^{-0.0110t}$ | 62.77 | 0.9884 | $C_t = 1.9363e^{-0.0119t}$ | 58.25 | 0.9843 | $C_t = 1.9968e^{-0.0116t}$ | 59.74 | 0.9878 |
| S4 | Alisol | $C_t = 1.8428e^{-0.0116t}$ | 59.74 | 0.9742 | $C_t = 1.9476e^{-0.0133t}$ | 52.10 | 0.9924 | $C_t = 1.7076e^{-0.0121t}$ | 57.27 | 0.9849 |
| S5 | Plinthosol | $C_t = 1.7637e^{-0.0104t}$ | 66.63 | 0.9650 | $C_t = 1.7864e^{-0.0126t}$ | 55.00 | 0.9655 | $C_t = 1.9725e^{-0.0121t}$ | 57.27 | 0.9923 |




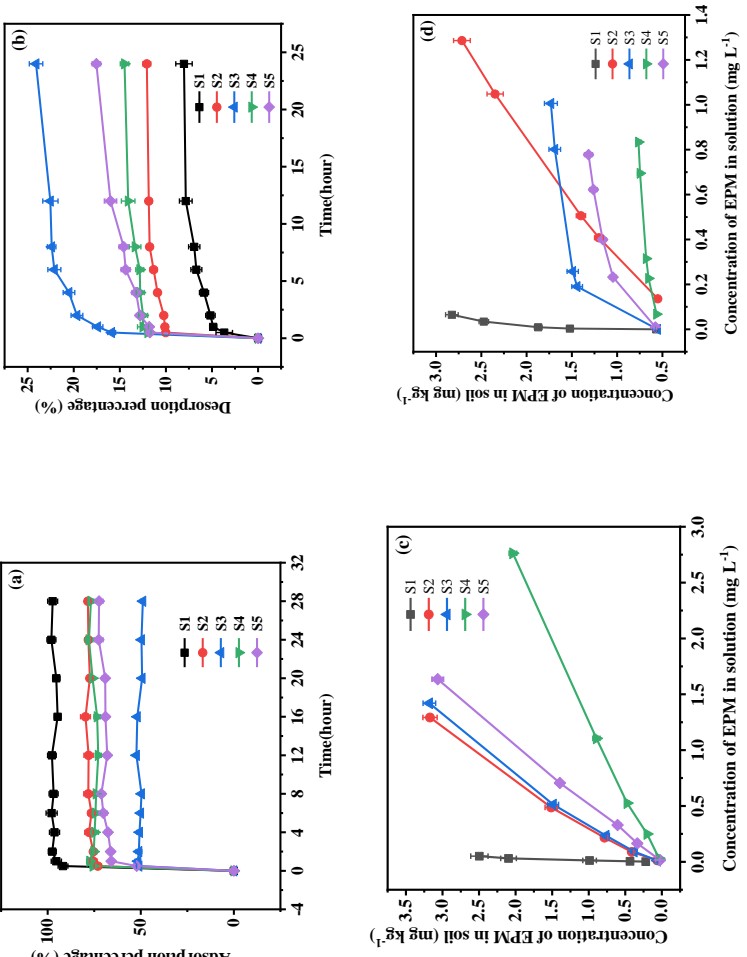

**Fig. 1** Adsorption (a) and desorption (b) kinetic curves and Adsorption (c) and desorption (d) isothermal curves of EPM in five different agricultural soils (S1 to S5 are defined in Table 1). Values are the means ± standard error (n=3).







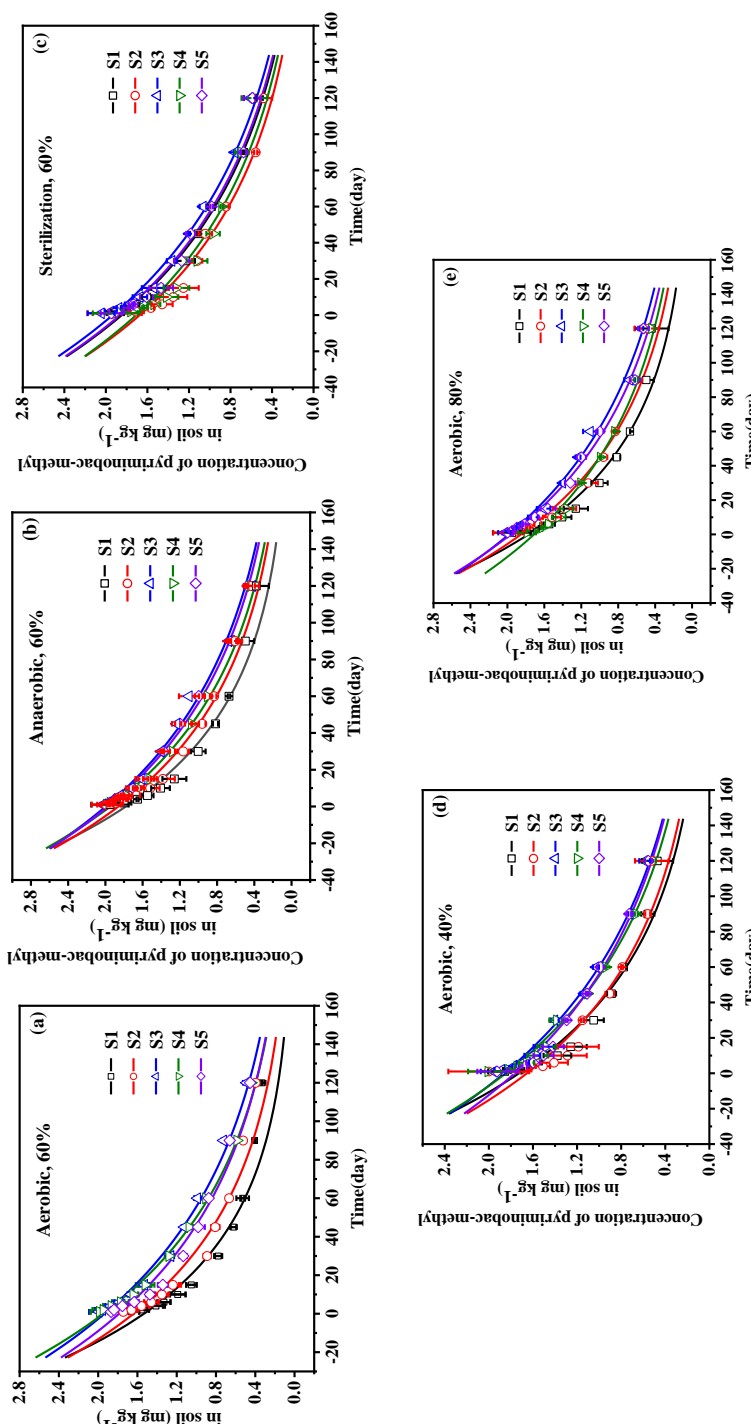

**Fig. 2** Degradation kinetics of EPM under aerobic (a), anaerobic (b), sterilization (c) conditions with 60% moisture, under aerobic conditions with 40% moisture(d) and with 80% moisture(e)in five different agricultural soils (S1 to S5 are defined in Table 1). Values are the means ± standard error (n=3).







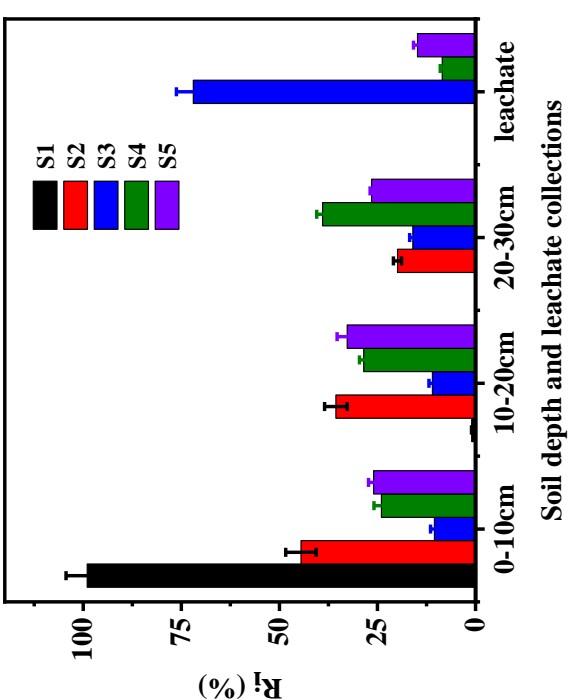

**Fig. 3** Distribution of EPM in soil column and leachate of five different agricultural soils (S1 to S5 are defined in Table 1)




