# Peer review of "Environmental behaviors of (*E*)-Pyriminobac-methyl in agricultural soils"

_SOIL, 2021_

## Author Comment (AC1)

**Soil**

**Manuscript No.:** SOIL-2021-103

**Manuscript Title:** Environmental behaviors of (*E*)-Pyriminobac-methyl in agricultural soils

**Article Type:** Research paper

**Authors:** Wenwen Zhou, Haoran Jia, Lang Liu, Baotong Li, Yuqi Li, Meizhu Gao

**Response to the first reviewer's comments**

**First of all, we would like to thank you for your valuable comments and suggestions which help us to improve our manuscript. Below we try to address all the points which you have indicated in your assessment opinions.**

**General comment:**

*Comment:* Pesticides, as chemical compounds widely and excessively used in the world, pose a significant threat to soil and water ecosystems. The presented publication raises the important issue of pesticides behavior in soil and their leaching potential. The manuscript is generally well written and contains many research results, however some issues that need to be improved. The introduction and discussion needs enhancement in some paragraphs and the figures should be corrected as they are illegible. All recommendations are listed in the below comments.

*Response:* **Thank you very much for your support of our manuscript. We further revised our manuscript according to your comments.**

**Specific comments:**

*Comment 1:* Abstract:

Resents well-organized information reflecting the contents of the manuscript.

*Response 1:* **This suggestion has been adopted. We have revised the abstract as follows,** '(*E*)-Pyriminobac-methyl (EPM), a pyrimidine benzoic acid esters herbicide, has a high potential as weedicide; nevertheless, its environmental behaviors are still not well understood. In this study, we systematically investigated for the first time the adsorption–desorption, degradation, and leaching behaviors of EPM in agricultural soils from five exemplar sites in China (Phaeozem: S1, Anthrosol: S2, Ferralsol: S3, Alisol: S4, and Plinthosol: S5) through laboratory simulation experiments. Our results show that the EPM adsorption–desorption results were well fitted by the Freundlich model ($R^2 > 0.9999$). In the analyzed soils, the Freundlich adsorption (i.e., $K_{f\text{-}ads}$) and desorption (i.e., $K_{f\text{-}des}$) coefficients of EPM varied between 0.85–32.22 $mg^{1-1/n} L^{1/n} kg^{-1}$ and between 0.78–5.02 $mg^{1-1/n} L^{1/n} kg^{-1}$, respectively. The mobility of EPM in the soils S1–S5 was categorized as immobile, slightly mobile, highly mobile, slightly mobile, and slightly mobile, respectively. Moreover, the degradation of EPM reflected first-order kinetics: its half-life ranged between 37.46–66.00 d depending on the environmental conditions, and abiotic degradation was predominant in the degradation

of this compound. Overall, the high leaching ability and desorption capacity of EPM were accompanied by a low adsorption capacity and there were no significant relationships between pH and the leaching rate of EPM in the five types of soils. In contrast, the organic matter content, cation exchange capacity, and soil clay content were the main responsible for the observed leaching rates. We found that EPM degrades easily, has a high adsorption affinity and a low mobility in S1, which result in a low contamination risk for groundwater systems. On the contrary, this compound degrades slowly in S2, S3, S4, and S5, due to a low adsorption affinity and moderate mobility, which result in a high contamination risk for groundwater systems. Therefore, our results may serve as a reference for evaluating the risks involved in the increasingly wide application of this compound.'

**Comment 2:** Keywords:

Should not be included in the title. Please reworded.

*Response 2:* **This suggestion has been adopted. We have revised the keywords as follows,** '(*E*)-pyriminobac-methyl, herbicide, soil organic matter, $K_{OC}$, risk assessment '**.**

**Comment 3:** Introduction:

l.34-46 What are the national standards/regulations for herbicide use in China and what are the detected exceeding of their concentrations?

*Response 3:* **This suggestion has been adopted.** China has the National food safety standard --- Maximum residue limits for pesticides in food (Gb, 2021), which contains

the maximum residual limit (MRL) and acceptable daily intakes (ADI) of 548 commonly used pesticides, for example, the MRL of pyriminobac-methyl in paddy rice and brown rice is 0.2 and 0.1 PPM respectively, and the ADI of pyriminobac-methyl is 0.02 PPM.

Most studies have reported that with the increasing use of glyphosate (a non-selective herbicide), especially in tea plantations (the detected MRL of glyphosate is 4.12 PPM much bigger than the limited value 1 PPM) and aquatic systems, the problem of excessive residues of glyphosate has attracted more and more attention, raising potential environmental threats and public health concerns(Liu et al., 2021; Luo et al., 2019; Huang et al., 2016).

**Reference:**

GB: National food safety standard — Maximum residue limits for pesticides in food, 2021.

Huang, J. L., Xiu-Ying, L. I., Lin, S. Y., Guo, X. D., and Quality: Determination of glyphosate residues in tea by ion chromatography, Journal of Food Safety and Quality, 2016.

Liu, J., Dong, C., Zhai, Z., Tang, L., and Wang, L.: Glyphosate-induced lipid metabolism disorder contributes to hepatotoxicity in juvenile common carp, Environmental Pollution, 269, 116186, 2021.

Luo, F. M., Wu, X. D., and liu, X. Y.: Determination of Pu'er Tea by High Performance Liquid Chromatography Tandem Mass Spectrometry Uncertainty Evaluation of Glyphosate Residues, Analysis and Testing, 144-148, 2019.

_**Comment 4:**_ l.68: Double parenthesis. Please correct

_**Response 4:**_ **This suggestion has been adopted.** We have corrected.

***Comment 5:*** l.69: Please explain the acronym 'PM'

***Response 5:*** **This suggestion has been adopted.** 'PM' is the abbreviated form of 'Pyriminobac-methyl'.

***Comment 6:*** l.75-83: What is the greater risk - leaching or uptake by plants? How half-life time of EPM affects the residence time of a compound in soil. Please outline the background for the research.

***Response 6:*** **This suggestion has been adopted. We deem that leaching of herbicide is more harmful than uptake by plants.** Indeed, the harm of weedicide leaching have been frequently reported in groundwaters. Several studies have indicated that the leaching risk potential of herbicides to groundwater is positively correlated with its mobility in soil (Bernard et al., 2005; Nofziger et al., 2015; Andrews and Trickett, 2010; Chen et al., 2021; Wang et al., 2019; Silva et al., 2019; Kaur et al., 2021; Willett et al., 2020). In the work of Paz and Rubio (2006), the inappropriate use of herbicides could result in the contamination of groundwater were reported, for instance, terbumeton, bromacil, and simazine herbicides have the highest risk of leaching because of their high mobility and low $K_{oc}$ (32–158 mg $L^{-1}$). The remaining herbicides are strongly adsorbed by clay particles and organic matter, thus minimising the risk of leaching through the soil profile and into groundwater. Guimares et al. (2019), who found that hexazinone (herbicide) proved to be a potential contaminant of groundwater and metribuzin (herbicide) presented high leaching in the soil profile. As well as metribuzin,

atrazine was found to be accumulated in algal cells, which indicates that herbicide pollution might eventually affect the marine food web and even threaten the seafood safety of human beings (Yang et al., 2019).

On the other hand, Kolakowski et al. (2020) and Mehdizadeh et al. (2021) reported that the residue levels of herbicides which were uptake by plants and the risk to consumers depends on the application technique, the environmental conditions, the stage of growth of plants, the volume of use, water quality and the use of coadjuvants. They also found that glyphosate (a most widely used chemical herbicide) is safe because neither glyphosate nor its primary degradation product, aminomethylphosphonic acid (AMPA), was associated with any known human health effects. Among other factors, human exposure through the diet is low because of low residue levels in foods and glyphosate disrupts an enzymatic pathway which exists in plants and fungi but not in animals or humans. EPM is also proved to be safe in rice. Jia et al. (2020) showed that the detected MRL of EPM in paddy rice is 0.0092 PPM far less than the limited value 0.2 PPM. Hence, previous knowledge of the physico-chemical properties of soils cultivated with crops is essential to recommend the use of these herbicides in weed management.

In the paddy rice field, the half-life of EPM calculated from 4.0 to 19.3 days (half-life $\leqslant$ 30 day, easily degradable) (Gb, 2014a) monitored in the Lake Biwa basin, Japan (Iwafune et al., 2010), the sorption constants of the OC ($K_{oc}$) values ranged from 372 to 741 ( $200 < K_{oc} \leqslant 1000$, sub-difficultly adsorbed compound)(Gb, 2014b)

conducted with Habikino and Ushiku soils in Japan(Inao et al., 2009), indicating that EPM is low-persistence herbicide, which result in a low contamination risk for groundwater systems. The Japanese Environment Agency sets limits for residues in paddy rice discharge water by allowing for a 10-fold dilution in river water and applying the drinking-water limit of EPM is 200 $\mu$g L$^{-1}$ (Hamilton et al., 2003). In China, EPM has been registered to control grassy weeds in paddy rice and brown rice fields at present (Gb, 2021).

**Reference:**

Andrews, J. and Trickett, S.: Future herbicide use is threatened by leaching, Farmers Weekly, 2010.

Bernard, H., Chabalier, P. F., Chopart, J. L., Legube, B., and Vauclin, M.: Assessment of Herbicide Leaching Risk in Two Tropical Soils of Reunion Island (France), 34, 534-543, https://doi.org/10.2134/jeq2005.0534, 2005.

Chen, Y., Han, J., Chen, D., Liu, Z., Zhang, K., and Hu, D.: Persistence, mobility, and leaching risk of flumioxazin in four Chinese soils, Journal of Soils and Sediments, 21, 1743-1754, 10.1007/s11368-021-02904-3, 2021.

GB: Test Guidelines of the Environmental Safety Assessment for Chemical Pesticides-Part 1: Transformation in Soils, 2014a.

GB: Test Guidelines of the Environmental Safety Assessment for Chemical Pesticides-Part 4: Adsorption/Desorption in Soils, 2014b.

Hamilton, D. J., Ambrus, Á., Dieterle, R. M., Felsot, A. S., Harris, C. A., Holland, P. T., Katayama, A., Kurihara, N., Linders, J., Unsworth, J., and Wong, S.-S.: Regulatory limits for pesticide residues in water (IUPAC Technical Report) Pure and Applied Chemistry, 75, 1123-1155, doi:10.1351/pac200375081123, 2003.

Inao, K., Mizutani, H., Yogo, Y., and Ikeda, M.: Improved PADDY model including

photoisomerization and metabolic pathways for predicting pesticide behavior in paddy fields: Application to the herbicide pyriminobac-methyl, Journal of Pesticide Science, 34, 273-282, 10.1584/jpestics.G09-20, 2009.

Iwafune, T., Inao, K., Horio, T., Iwasaki, N., Yokoyama, A., and Nagai, T.: Behavior of paddy pesticides and major metabolites in the Sakura River, Ibaraki, Japan, Journal of Pesticide Science, advpub, 1001130109-1001130109, 10.1584/jpestics.G09-49, 2010.

Jia, H. R., Zhang, Y., Li, W., Li, B. T., and Zhou, W. W.: Residue dynamics and dietary risk assessment of pyriminobac-methyl in rice, Acta Scientiae Circumstantiae, 4, 1491-1499, 2020.

National Health Commission, P.: National food safety standard — Maximum residue limits for pesticides in food, 2021.

Nofziger, D. L., Chen, J. S., Hornsby, A. G., and Sons, L.: Uncertainty in Pesticide Leaching Risk Due to Soil Variability, John Wiley, 2015.

Paz, J. and Rubio, J. L.: Application of a GIS–AF/RF model to assess the risk of herbicide leaching in a citrus-growing area of the Valencia Community, Spain, Science of the Total Environment, 371, 44-54, 2006.

Silva, T. S., de Freitas Souza, M., Maria da Silva Teófilo, T., Silva dos Santos, M., Formiga Porto, M. A., Martins Souza, C. M., Barbosa dos Santos, J., and Silva, D. V.: Use of neural networks to estimate the sorption and desorption coefficients of herbicides: A case study of diuron, hexazinone, and sulfometuron-methyl in Brazil, Chemosphere, 236, 124333, https://doi.org/10.1016/j.chemosphere.2019.07.064, 2019.

Wang, W., Liang, Y., Yang, J., Tang, G., Zhou, Z., Tang, R., Dong, H., Li, J., and Cao, Y.: Ionic Liquid Forms of Mesotrione with Enhanced Stability and Reduced Leaching Risk, ACS Sustainable Chemistry & Engineering, 7, 16620-16628, 10.1021/acssuschemeng.9b03948, 2019.

Willett, C. D., Grantz, E. M., Sena, M. G., Lee, J. A., Brye, K. R., and Clarke, J. A.: Soil sorption characteristics of benzobicyclon hydrolysate and estimated leaching risk in soils used for rice production, Environmental Chemistry, 17, 445-456, https://doi.org/10.1071/EN19189, 2020.

***Comment 7:*** l.109-115: Is the method used 'own' or standardized? The individual

analytical steps indicate the determination of the available EPM fraction, not the total

fraction (usually used with more aggressive / stronger solvents).

*Response 7:* **This suggestion has been adopted.** The method used 'own'. The details

of the extraction method and HPLC-MS analytical method were reported previously (Jia

et al., 2019a).

The recovery of EPM from paddy water investigated QuEChERS using five different

solvents for extraction: methyl alcohol, acetonitrile, dichloromethane, acetone, and

Ethyl acetate. The results showed that acetonitrile extraction recovery was the highest

among the five solvents(Jia et al., 2019b).

**Reference:**

Jia, H. R., Zhang, Y., Li, W., and Li, B. T.: HPLC- tandem Mass Spectrometry Method for the
Determination of Pyriminobac- methyl 10% WP, Agrochemicals, 58, 106-108, 2019a.

Jia, H. R., Zhang, Y., W, L., Li, B. T., Shi, X. G., and Tang, L. M.: Residue of pyriminobac-methyl in
rice and environment, Chinese Journal of Pesticide Science, 2, 250-254, 2019b.

*Comment 8:* l.125: Please provide the determination parameters of the chromatographic

method, i.e. repeatability, reproducibility, recovery, measurement uncertainty, detection

limit and limit of quantification.

*Response 8:* **This suggestion has been adopted.** The results of linearity, LOD, LOQ,

and matrix effect are summarized in Table 1. The calibration curve of EPM (0.005, 0.01,

0.05, 0.1, 0.5, and 1mg $kg^{-1}$ ) showed a high correlation coefficient ($R^2 > 0.994$) in all

matrices. To evaluate its specificity, the method was applied to blank samples of different matrices. No interference was detected during the retention time. The LODs of EPM ranged from 0.001 to 0.013 mg kg$^{-1}$ , while the LOQs were 0.004 – 0.049 mg kg$^{-1}$ for all samples. All matrices had moderate matrix effects (ME < 10%), thus, the matrix standard curve could be ignored (Li et al., 2019).

Five parallel tests were conducted for each matrix spiked with EPM at three different levels (0.005, 0.01, and 0.1 mg kg$^{-1}$). After sample pretreatment by the optimized QuEChERS procedure, the recovery of EPM in the various matrices ranged between 90.95% and 110.12%, with RSDs of 1.3% – 9.8% for repeatability (Table 2), and with RSDs of 3.63% – 8.49% for repeatability (Table 3). Thus, the developed analytical method fulfills the requirements of SANTE/11813/2017 guidelines and fall within the range of 70–120% for recovery and less than 20% for RSD (Sante, 2017).

Table 1 Linear regression equation, limit of detection (LOD), limit of quantification (LOQ), and matrix effect for EPM in various matrices

| Matrixes | Linear Equation | $R^2$ | LOD (mg kg$^{-1}$) | LOQ (mg kg$^{-1}$) | Matrix effect (%) |
|---|---|---|---|---|---|
| Acetonitrile | y = 254868 x +2080.3 | 0.9999 | - | - | - |
| Paddy water | y = 188630 x + 3059.2 | 0.9983 | 0.012 | 0.045 | 1.84 |
| Paddy soil | y = 172440 x + 3779.2 | 0.9961 | 0.002 | 0.008 | 1.86 |
| Rice straw | y = 141678 x + 2501.3 | 0.9949 | 0.001 | 0.004 | 1.36 |
| Brown rice | y = 138732 x + 492.6 | 0.9985 | 0.011 | 0.042 | 1.01 |
| Rice Hulls | y = 152673 x + 1517.1 | 0.9991 | 0.013 | 0.049 | 1.48 |

Table 2 Recovery and relative standard deviation (RSD) of EPM in various matrices spiked at levels of 0.005, 0.01, and 0.1 mg kg$^{-1}$ (n=5)

| Matrix | Spiked level (mg kg$^{-1}$) | Recovery(%) | | | | | Mean recovery (%) | RSD (%) |
|---|---|---|---|---|---|---|---|---|
| | | 1 | 2 | 3 | 4 | 5 | | |
| Paddy water | 0.005 | 93.51 | 107.03 | 91.03 | 90.95 | 104.35 | 97.37 | 7.93 |
| | 0.01 | 91.96 | 96.10 | 97.76 | 101.58 | 110.12 | 99.50 | 6.90 |
| | 0.1 | 93.93 | 92.88 | 103.45 | 108.15 | 109.67 | 101.62 | 7.72 |
| Paddy soil | 0.005 | 104.94 | 99.57 | 100.35 | 102.84 | 102.35 | 102.01 | 2.09 |
| | 0.01 | 108.01 | 93.85 | 94.10 | 94.22 | 104.79 | 98.99 | 6.93 |
| | 0.1 | 98.22 | 102.26 | 108.82 | 97.82 | 97.26 | 100.88 | 4.82 |
| Rice straw | 0.005 | 100.73 | 109.29 | 91.89 | 93.56 | 108.38 | 100.77 | 8.02 |
| | 0.01 | 102.67 | 95.22 | 93.22 | 103.30 | 102.91 | 99.46 | 4.87 |
| | 0.1 | 93.09 | 95.27 | 109.84 | 90.19 | 95.39 | 96.76 | 7.87 |
| Brown rice | 0.005 | 91.10 | 91.72 | 104.98 | 107.54 | 104.22 | 99.91 | 7.87 |
| | 0.01 | 92.17 | 109.97 | 108.62 | 91.22 | 97.30 | 99.86 | 8.95 |
| | 0.1 | 108.94 | 92.88 | 91.52 | 95.18 | 95.98 | 96.90 | 7.18 |
| Rice Hulls | 0.005 | 100.09 | 98.40 | 104.96 | 105.42 | 97.88 | 101.35 | 3.55 |
| | 0.01 | 97.21 | 102.68 | 94.47 | 96.10 | 92.12 | 96.52 | 4.09 |
| | 0.1 | 100.09 | 92.77 | 93.50 | 93.92 | 98.87 | 95.83 | 3.53 |

Table 3 Reproducibility of the rention time, precursor signal, and retention factor of EPM

| | RSD (%) | |
|---|---|---|
| | Column-to-column reproducibility on five columns | Batch-to-batch reproducibility on six batches |
| Rention time | 4.89 | 6.01 |
| Precursor signal | 7.27 | 8.49 |
| Retention factor | 3.63 | 5.87 |

**Reference:**

Li, W., Zhang, Y., Jia, H., Zhou, W., Li, B., and Huang, H.: Residue analysis of tetraniliprole in rice and related environmental samples by HPLC/MS, Microchemical Journal, 150, 104168, https://doi.org/10.1016/j.microc.2019.104168, 2019.

SANTE: Guidance document on analytical quality control and method validation procedures for pesticide residues analysis in food and feed, 2017.

**Comment 9:** l.126-134: what is the estimated environmental stability of EPM based on the literature data? This information is necessary for appropriate analysis of the presented data.

*Response 9:* **This suggestion has been adopted.** In the paddy rice field, the half-life of EPM calculated from 4.0 to 19.3 days (half-life $\leqslant$ 30 day, easily degradable) (Gb, 2014a) monitored in the Lake Biwa basin, Japan (Iwafune et al., 2010), the sorption constants of the OC ($K_{oc}$) values ranged from 372 to 741 ( $200 < K_{oc} \leqslant 1000$, sub-difficultly adsorbed compound)(Gb, 2014b) conducted with Habikino and Ushiku soils in Japan(Inao et al., 2009), indicating that EPM is low-persistence herbicide, which result in a low contamination risk for groundwater systems.

**Reference:**

GB: Test Guidelines of the Environmental Safety Assessment for Chemical Pesticides-Part 4: Adsorption/Desorption in Soils, 2014b.

Inao, K., Mizutani, H., Yogo, Y., and Ikeda, M.: Improved PADDY model including photoisomerization and metabolic pathways for predicting pesticide behavior in paddy fields: Application to the herbicide pyriminobac-methyl, Journal of Pesticide Science, 34, 273-282, 10.1584/jpestics.G09-20, 2009.

Iwafune, T., Inao, K., Horio, T., Iwasaki, N., Yokoyama, A., and Nagai, T.: Behavior of paddy pesticides and major metabolites in the Sakura River, Ibaraki, Japan, Journal of Pesticide Science, advpub, 1001130109-1001130109, 10.1584/jpestics.G09-49, 2010.

**Comment 10:** l.218, 223, 230, 234: Illegible chart. Please correct

*Response 10:* **This suggestion has been adopted.** We have corrected the illegible

charts.

*__Comment 11:__* l.261-266: How were outliers dealt with?

*__Response 11:__* **This suggestion has been adopted.** Aguinis et al. (2013) recommendations on how to define, identify, and handle outliers are based on two overarching principles. The first category consists of error outliers, or data points that lie at a distance from other data points because they are the result of inaccuracies. The second category represents interesting outliers, which are accurate data points that lie at a distance from other data pointsand may contain valuable or unexpected knowledge. The third category refers to influential outliers, which are accurate data points that lie at a distance from other data points, are not error or interesting outliers, and also affect substantive conclusions. The approaches to identifying and handling error and interesting outliers are similar across data-analytic techniques. However, the way influential outliers are identified and handled depends on the particular technique-for example, regression versus SEM. Thus, they first provide a discussion regarding error and interesting outliers and then offer a separate treatment of influential outliers within each of the specific contexts of regression, SEM, and multilevel modeling. As seen in Figure 1, they recommendation is that all empirical studies follow the same sequence of steps. **We handled outliers as they recommend.**

[Figure]

**Figure 1. Decision-making tree summarizing sequential steps in the process of understanding the possible presence of outliers (Aguinis et al., 2013).**

[Figure]

**Figure 2. Decision-making tree summarizing sequential steps in the process of defining, identifying, and handling influential outliers in the context of regression, structural equation modeling (SEM), and multilevel modeling (Aguinis et al., 2013).**

**Reference:**

Aguinis, H., Gottfredson, R. K., and Joo, H.: Best-Practice Recommendations for Defining, Identifying, and Handling Outliers, Organizational Research Methods, 16, 270-301, 10.1177/1094428112470848, 2013.

*Comment 12:* Results and discussion:

Results are very good described but discussion should be strengthened especially in the first subsection. l.286-287: The logical statement. Please expand the discussion about sorption affinity.

*Response 12:* **This suggestion has been adopted.** The tendency of sorbed hydrophobic organic pollutant to become more strongly bound with increasing organic matter (OM) contents of the soils is well demonstrated for EPM. This is consistent with previously reported observations that for sorbents with organic-carbon content greater than 0.1%, a highly significant positively correlation was found between the adsorption constants of non-polar or weakly polar organic compounds and the OM of soils (Schwarzenbach and Westall, 1981; Chefetz et al., 2004). The main reason is that the OM of soils has special binding sites with organic pesticide molecules. With the increase of OM content, the adsorption sites also increased, thus increasing the herbicide adsorption capacity (Stevenson, 1972; Ahmad et al., 2001; Delle Site, 2001; Chianese et al., 2020). The role of different components of the OM in determining herbicide sorption has been clearly observed in previous studies. Hartley (1960) had speculated that the 'oily' constituent of the OM might be responsible for uptake of nonionic compounds by soil. The existence of such a lipid phase was supported by Schnitzer and Khan (1972), who reported the presence off attyacids and alkanes at the surface of the OM resulting from long alkyl chains projecting from the surface. They suggested that interactions such as hydrogen bondingmight be important in uptake of nonionic contaminants by this lipid fraction.

The hydrophobicity of the OM has generally been reported to originate from aromatic and alkyl domains of the organic matter component (Ahmad et al., 2001). Murphy and Zachara (1995) suggested the presence of heterogeneous sorption sites on the OM and considered the most hydrophobic domains as the most energetic and strong binding sites.

**Reference:**

Ahmad, R., Kookana, R. S., Alston, A. M., and Skjemstad, J. O.: The Nature of Soil Organic Matter Affects Sorption of Pesticides. 1. Relationships with Carbon Chemistry as Determined by [13]C CPMAS NMR Spectroscopy, Environmental Science & Technology, 35, 878-884, 10.1021/es001446i, 2001.

Chefetz, B., Bilkis, Y. I., and Polubesova, T.: Sorption–desorption behavior of triazine and phenylurea herbicides in Kishon river sediments, Water Research, 38, 4383-4394, https://doi.org/10.1016/j.watres.2004.08.023, 2004.

Chianese, S., Fenti, A., Iovino, P., Musmarra, D., and Salvestrini, S.: Sorption of Organic Pollutants by Humic Acids: A Review, 25, 918, 2020.

Delle Site, A.: Factors Affecting Sorption of Organic Compounds in Natural Sorbent/Water Systems and Sorption Coefficients for Selected Pollutants. A Review, Journal of Physical and Chemical Reference Data, 30, 187-439, 10.1063/1.1347984, 2001.

Hartley, G. S.: In Herbicides and the Soil;Woodford, Blackwell Scientific Publishing Company, Oxford1960.

Murphy, E. M. and Zachara, J. M.: The role of sorbed humic substances on the distribution of organic and inorganic contaminants in groundwater, Geoderma, 67, 103-124, https://doi.org/10.1016/0016-7061(94)00055-F, 1995.

Schnitzer, M. and Khan, S. U.: In Humic Substances in the Environment, Marcel Dekker Inc, New York1972.

Schwarzenbach, R. P. and Westall, J.: Transport of nonpolar organic compounds from surface water to groundwater. Laboratory sorption studies, Environmental Science & Technology, 15, 1360-1367, 10.1021/es00093a009, 1981.

***Comment 13:*** l.405-406: Repetition. This information is contained in the materials and methods section.

***Response 13:*** **This suggestion has been adopted.** We have deleted the duplicate information.

***Comment 14:*** Conclusions:

Conclusions are properly written and do not require corrections.

***Response 14:*** **Thank you very much!**

**Thanks again for your kindly comments.**

---

## Author Comment (AC2)

**Soil**

**Manuscript No.:** SOIL-2021-103

**Manuscript Title:** Environmental behaviors of (*E*)-Pyriminobac-methyl in agricultural soils

**Article Type:** Research paper

**Authors:** Wenwen Zhou, Haoran Jia, Lang Liu, Baotong Li, Yuqi Li, Meizhu Gao

**Response to the second reviewer's comments**

**First of all, we would like to thank you for your valuable comments and suggestions which help us to improve our manuscript. Below we try to address all the points which you have indicated in your assessment opinions.**

**General comment:**

*Comment:* This study provide results on herbicide EPM behaviour in paddy soils. I think that these experimental data bay help understand this compound in risk assessment.

*Response:* **Thank you very much for your support of our manuscript. We further revised our manuscript according to your comments.**

**Specific comments:**

*Comment 1:* Suggeestions:

For degradation study, did the author test the degradation products by MASS or other detection means?

*Response 1:* **This suggestion has been adopted.** We apologize for not analyzing and testing the degradation products. Thus, this experiment has been included in our work this year. Thank you for your valuable suggestions to improve our research.

*Comment 2:* It's recommended to provide the analytical method performance in validation, and typical chromatograms.

*Response 2:* **This suggestion has been adopted. We supplemented typical chromatograms of the analytical method performance.** The selective ion chromatograms of EPM in acetonitrile, paddy water, paddy soil, paddy straw, brown rice and rice hulls samples spiked at 0 and 0.1 mg kg$^{-1}$ were shown in Figure 1 (A-F). Five parallel tests were conducted for each matrix spiked with EPM at three different levels (0.005, 0.01, and 0.1 mg kg$^{-1}$). After sample pretreatment by the optimized QuEChERS procedure, the recovery of EPM in the various matrices ranged between 90.95% and 110.12%, with RSDs of 1.3% – 9.8% for repeatability (Table 1), and with RSDs of 3.63% – 8.49% for repeatability (Table 2). Five parallel tests were conducted for the blank matrix of paddy water samples spiked at 0.005, 0.01 and 0.1 mg kg$^{-1}$ of EPM, respectively, and the chromatograms were shown in Figure 2 (A-C).

Chromatograms of EPM on the five columns of one batche and on the five columns of the different batches were shown in Figure 3 (A-B). Thus, the developed analytical method fulfills the requirements of SANTE/11813/2017 guidelines and fall within the range of 70 – 120 % for recovery and less than 20% for RSD (Sante, 2017).

[Figure]

Figure 1   The selective ion chromatograms of blank (A) acetonitrile, (B) paddy water, (C) paddy soil, (D) paddy straw, (E) brown rice and (F) rice hulls samples spiked at 0 and 0.1 mg kg$^{-1}$.

Table 1 Recovery and relative standard deviation (RSD) of EPM in various matrices spiked at levels of 0.005, 0.01, and 0.1 mg kg$^{-1}$ (n=5)

| Matrix | Spiked level | Recovery(%) | | | | | Mean recovery | RSD |
|---|---|---|---|---|---|---|---|---|
| | (mg kg$^{-1}$) | 1 | 2 | 3 | 4 | 5 | (%) | (%) |
| Paddy water | 0.005 | 93.51 | 107.03 | 91.03 | 90.95 | 104.35 | 97.37 | 7.93 |
| | 0.01 | 91.96 | 96.10 | 97.76 | 101.58 | 110.12 | 99.50 | 6.90 |
| | 0.1 | 93.93 | 92.88 | 103.45 | 108.15 | 109.67 | 101.62 | 7.72 |
| Paddy soil | 0.005 | 104.94 | 99.57 | 100.35 | 102.84 | 102.35 | 102.01 | 2.09 |
| | 0.01 | 108.01 | 93.85 | 94.10 | 94.22 | 104.79 | 98.99 | 6.93 |
| | 0.1 | 98.22 | 102.26 | 108.82 | 97.82 | 97.26 | 100.88 | 4.82 |
| Rice straw | 0.005 | 100.73 | 109.29 | 91.89 | 93.56 | 108.38 | 100.77 | 8.02 |
| | 0.01 | 102.67 | 95.22 | 93.22 | 103.30 | 102.91 | 99.46 | 4.87 |
| | 0.1 | 93.09 | 95.27 | 109.84 | 90.19 | 95.39 | 96.76 | 7.87 |
| Brown rice | 0.005 | 91.10 | 91.72 | 104.98 | 107.54 | 104.22 | 99.91 | 7.87 |
| | 0.01 | 92.17 | 109.97 | 108.62 | 91.22 | 97.30 | 99.86 | 8.95 |
| | 0.1 | 108.94 | 92.88 | 91.52 | 95.18 | 95.98 | 96.90 | 7.18 |
| Rice Hulls | 0.005 | 100.09 | 98.40 | 104.96 | 105.42 | 97.88 | 101.35 | 3.55 |
| | 0.01 | 97.21 | 102.68 | 94.47 | 96.10 | 92.12 | 96.52 | 4.09 |
| | 0.1 | 100.09 | 92.77 | 93.50 | 93.92 | 98.87 | 95.83 | 3.53 |

[Figure]

Figure 2    Recovery of EPM in paddy water spiked at levels of 0.005, 0.01, and 0.1

mg kg$^{-1}$ (n=5)

Table 2 Reproducibility of the rention time, precursor signal, and retention factor of EPM

| | RSD (%) | |
| --- | --- | --- |
| | Column-to-column reproducibility on five columns | Batch-to-batch reproducibility on six batches |
| Rention time | 4.89 | 6.01 |
| Precursor signal | 7.27 | 8.49 |
| Retention factor | 3.63 | 5.87 |

[Figure]

Figure 3 Chromatograms of EPM in paddy water on the five columns of the first batche (A) and on the five columns of the different batches (B) samples spiked at 0.1 mg kg$^{-1}$

**Reference:**

SANTE: Guidance document on analytical quality control and method validation procedures for pesticide residues analysis in food and feed, 2017.

**Thanks again for your kindly comments.**

---

## Author Response (AR1)

**Soil**

**Manuscript No.:** SOIL-2021-103

**Manuscript Title:** Environmental behaviors of (*E*)-Pyriminobac-methyl in agricultural soils

**Article Type:** Research paper

**Authors:** Wenwen Zhou, Haoran Jia, Lang Liu, Baotong Li, Yuqi Li, Meizhu Gao

**Response to the first reviewer's comments**

**First of all, we would like to thank you for your valuable comments and suggestions which help us to improve our manuscript. Below we try to address all the points which you have indicated in your assessment opinions.**

**General comment:**

*Comment:* Pesticides, as chemical compounds widely and excessively used in the world, pose a significant threat to soil and water ecosystems. The presented publication raises the important issue of pesticides behavior in soil and their leaching potential. The manuscript is generally well written and contains many research results, however some issues that need to be improved. The introduction and discussion needs enhancement in some paragraphs and the figures should be corrected as they are illegible. All recommendations are listed in the below comments.

*Response:* **Thank you very much for your support of our manuscript. We further revised our manuscript according to your comments.**

**Specific comments:**

*Comment 1:* Abstract:

Resents well-organized information reflecting the contents of the manuscript.

*Response 1:* **This suggestion has been adopted.** We have revised the abstract. **Please see Lines 21-39 in the revised manuscript**.

*Comment 2:* Keywords:

Should not be included in the title. Please reworded.

*Response 2:* **This suggestion has been adopted.** We have revised the keywords. **Please see Line 40 in the revised manuscript**.

*Comment 3:* Introduction:

l.34-46 What are the national standards/regulations for herbicide use in China and what are the detected exceeding of their concentrations?

*Response 3:* **This suggestion has been adopted.** We have added the national standards for herbicide use in China and what are the detected exceeding of their concentrations. **Please see Lines 45-54 in the revised manuscript**.

*Comment 4:* l.68: Double parenthesis. Please correct

*Response 4:* **This suggestion has been adopted.** We have corrected. **Please see Line 66 in the revised manuscript**.

*Comment 5:* l.69: Please explain the acronym 'PM'

*Response 5:* **This suggestion has been adopted.** 'PM' is the abbreviated form of 'Pyriminobac-methyl'. **Please see Line 49 in the revised manuscript**.

*Comment 6:* l.75-83: What is the greater risk - leaching or uptake by plants? How half-life time of EPM affects the residence time of a compound in soil. Please outline the background for the research.

*Response 6:* **This suggestion has been adopted.** We deem that leaching of herbicide is more harmful than uptake by plants. **Please see Lines 86-99 in the revised manuscript**.

*Comment 7:* l.109-115: Is the method used 'own' or standardized? The individual analytical steps indicate the determination of the available EPM fraction, not the total fraction (usually used with more aggressive / stronger solvents).

*Response 7:* **This suggestion has been adopted.** The method used 'own'. The details of the extraction method and HPLC-MS analytical method were reported previously (Jia et al., 2019a).

The recovery of EPM from paddy water investigated QuEChERS using five different solvents for extraction: methyl alcohol, acetonitrile, dichloromethane, acetone, and Ethyl acetate. The results showed that acetonitrile extraction recovery was the highest among the five solvents(Jia et al., 2019b). **Please see Line 214 in the revised**

**manuscript**.

**Reference:**

Jia, H. R., Zhang, Y., Li, W., and Li, B. T.: HPLC- tandem Mass Spectrometry Method for the Determination of Pyriminobac- methyl 10% WP, Agrochemicals, 58, 106-108, 2019a.

Jia, H. R., Zhang, Y., W, L., Li, B. T., Shi, X. G., and Tang, L. M.: Residue of pyriminobac-methyl in rice and environment, Chinese Journal of Pesticide Science, 2, 250-254, 2019b.

*Comment 8:* l.125: Please provide the determination parameters of the chromatographic method, i.e. repeatability, reproducibility, recovery, measurement uncertainty, detection limit and limit of quantification.

*Response 8:* **This suggestion has been adopted.** The detail information of the determination parameters of the chromatographic method, i.e., repeatability, reproducibility, recovery, measurement uncertainty, detection limit and limit of quantification were shown in the supplementary material (Fig. S2-S4 and Table S2-S4). **Please see Fig. S2-S4 and Table S2-S4 in the revised supplementary material.**

*Comment 9:* l.126-134: what is the estimated environmental stability of EPM based on the literature data? This information is necessary for appropriate analysis of the presented data.

*Response 9:* **This suggestion has been adopted.** We have added the estimated environmental stability of EPM based on the literature data. **Please see Lines 100-107 in the revised manuscript**.

*Comment 10:* l.218, 223, 230, 234: Illegible chart. Please correct

*Response 10:* **This suggestion has been adopted.** We have corrected the illegible charts. **Please see Lines 250, 255, 262, 267 in the revised manuscript**.

*Comment 11:* l.261-266: How were outliers dealt with?

*Response 11:* **This suggestion has been adopted.** Aguinis et al. (2013) recommendations on how to define, identify, and handle outliers are based on two overarching principles. The first category consists of error outliers, or data points that lie at a distance from other data points because they are the result of inaccuracies. The second category represents interesting outliers, which are accurate data points that lie at a distance from other data pointsand may contain valuable or unexpected knowledge. The third category refers to influential outliers, which are accurate data points that lie at a distance from other data points, are not error or interesting outliers, and also affect substantive conclusions. The approaches to identifying and handling error and interesting outliers are similar across data-analytic techniques. However, the way influential outliers are identified and handled depends on the particular technique-for example, regression versus SEM. Thus, they first provide a discussion regarding error and interesting outliers and then offer a separate treatment of influential outliers within each of the specific contexts of regression, SEM, and multilevel modeling. As seen in Figure 1, they recommendation is that all empirical studies follow the same sequence of steps. **We handled outliers as they recommend.**

[Figure]

**Fig. 1 Decision-making tree summarizing sequential steps in the process of understanding the possible presence of outliers (Aguinis et al., 2013).**

[Figure]

**Fig. 2 Decision-making tree summarizing sequential steps in the process of defining, identifying, and handling influential outliers in the context of regression, structural equation modeling (SEM), and multilevel modeling (Aguinis et al., 2013).**

**Reference:**

Aguinis, H., Gottfredson, R. K., and Joo, H.: Best-Practice Recommendations for Defining, Identifying, and Handling Outliers, Organizational Research Methods, 16, 270-301, 10.1177/1094428112470848, 2013.

*Comment 12:* Results and discussion:

Results are very good described but discussion should be strengthened especially in the first subsection. l.286-287: The logical statement. Please expand the discussion about sorption affinity.

*Response 12:* **This suggestion has been adopted.** We have expanded the discussion about sorption affinity. **Please see Lines 317-332 in the revised manuscript**.

*Comment 13:* l.405-406: Repetition. This information is contained in the materials and methods section.

*Response 13:* **This suggestion has been adopted.** We have deleted the duplicate information.

*Comment 14:* Conclusions:

Conclusions are properly written and do not require corrections.

*Response 14:* **Thank you very much!**

**Thanks again for your kindly comments.**

**Response to the second reviewer's comments**

**First of all, we would like to thank you for your valuable comments and suggestions which help us to improve our manuscript. Below we try to address all the points which you have indicated in your assessment opinions.**

**General comment:**

***Comment***: This study provide results on herbicide EPM behaviour in paddy soils. I think that these experimental data bay help understand this compound in risk assessment.

***Response*: Thank you very much for your support of our manuscript. We further revised our manuscript according to your comments.**

**Specific comments:**

***Comment 1***: Suggeestions:

For degradation study, did the author test the degradation products by MASS or other detection means?

***Response 1*: This suggestion has been adopted.** We apologize for not analyzing and testing the degradation products. Thus, this experiment has been included in our work this year. Thank you for your valuable suggestions to improve our research.

***Comment 2:*** It's recommended to provide the analytical method performance in validation, and typical chromatograms.

***Response 2:*** **This suggestion has been adopted.** We supplemented typical chromatograms of the analytical method performance in the supplementary material (Fig. S2-S4). **Please see Fig. S2-S4 in the revised supplementary material.**

**Thanks again for your kindly comments.**

**Response to the third reviewer's comments**

**First of all, we would like to thank you for your valuable comments and suggestions which help us to improve our manuscript. Below we try to address all the points which you have indicated in your assessment opinions.**

**General comment:**

*Comment:* The manucript under the title "Environmental behaviors of (*E*)-Pyriminobac-methyl in agricultural soils" is relevant to the scope of the Journal, scientifically sound and valid. The Authors performed an immense work to study the adsorption–desorption, degradation, and leaching behaviors of EPM in physicochemically various soils, from five exemplar sites in China.

The data presented in the study is comprehensive, very detailed, discussed thoroughly an the conclusions supported by the mathematical models, which makes the paper a great source of data - also as an experimental approach in the pesticide studies.

The only suggestion I would have is to postpone part. 2.2 Extraction and final analyses a bit further in the Matrials and Methods part, as the potential reader may be confused by getting the detailed information about the extraction and final assessment of the pesticide before the methodological approach is revealed (description of the soil spiking, adsorption-desorption studies etc.). I believe that would make the work more transparent.

Differences in the sorptive behaviour of the soils are well explained (by various mineralogical composition of soils, especially clay minerals and organic matter contents, CEC value etc.). Results of this study demonstrate the high degradability of EPM, as well as its high adsorption affinity and low mobility in soils with abundant organic matter content and high cation exchange capacity.

The paper may serve as a a solid basis for predicting the environmental impacts of EPM and a great reference for the other researchers in this field, as there are still only a few studies on the EPM behaviour in the soil. Therefore, I support its publication.

*Response:* **Thank you very much for your support of our manuscript. We further revised our manuscript according to your comments.**

**Specific comments:**

*Comment 1:* The only suggestion I would have is to postpone part. 2.2 Extraction and final analyses a bit further in the Matrials and Methods part, as the potential reader may be confused by getting the detailed information about the extraction and final assessment of the pesticide before the methodological approach is revealed (description of the soil spiking, adsorption-desorption studies etc.). I believe that would make the work more transparent.

*Response 1:* **This suggestion has been adopted.** We have moved part 2.2. Extraction and final analyses after part 2.5. Leaching experiments. Thank you for your valuable suggestions to improve our research. **Please see Lines 208-236 in the revised**

**manuscript**.

**Thanks again for your kindly comments.**